# Objective Biomarkers of Outdoor Activity (Vitamin D and CUVAF) in Young Adults with Myopia During and After the SARS-CoV-2 Pandemic

**DOI:** 10.3390/biomedicines13082042

**Published:** 2025-08-21

**Authors:** Natali Gutierrez-Rodriguez, Miriam de la Puente-Carabot, Javier Andres Rodriguez-Hilarion, Jorge A. Ramos-Castaneda, Valentina Bilbao-Malavé, Carlos Javier Avendaño-Vasquez, Jorge Gonzalez-Zamora, Sandra Johanna Garzón-Parra, Sergio Recalde

**Affiliations:** 1Optometry Research Group, Optometry School, Universidad Antonio Nariño, Bogota 110221, Colombia; nataligutierrez@uan.edu.co (N.G.-R.); sjgarzonp@unal.edu.co (S.J.G.-P.); 2Department of Ophthalmology, Clínica Universidad de Navarra, 31008 Pamplona, Spain; mdelapuentec@unav.es (M.d.l.P.-C.); jgz.oftalmo@gmail.com (J.G.-Z.); 3Retinal Pathologies and New Therapies Group, Experimental Ophthalmology Laboratory, Department of Ophthalmology, Universidad de Navarra, 31008 Pamplona, Spain; valentina@clinic.cat; 4Research Group Innovacion y Cuidado, Faculty of Nursing, Universidad Antonio Nariño, Bogota 110221, Colombia; jhilarion@uan.edu.co (J.A.R.-H.); joramos98@uan.edu.co (J.A.R.-C.); javierunvasquez76@gmail.com (C.J.A.-V.); 5Department of Ophthalmology, Bellvitge University Hospital, 08907 Barcelona, Spain; 6Navarra Institute for Health Research (IdiSNA), 31008 Pamplona, Spain

**Keywords:** myopia, biomarker, vitamin D, CUVAF, SARS, pandemic

## Abstract

**Background/Objectives**: Intrinsic biomarkers, such as serum vitamin D levels and the conjunctival ultraviolet autofluorescence (CUVAF) area, have been proposed to quantify sunlight exposure. Evidence suggests that reduced outdoor activity during the SARS-CoV-2 pandemic accelerated the progression of myopia; however, there is little information on the impact of such restrictions on vitamin D levels and CUVAF area in populations with myopia. This study aims to assess the association between serum vitamin D levels and conjunctival ultraviolet autofluorescence area (CUVAF) in young adults with myopia during and after the pandemic, as well as its relationship with sun exposure habits and the use of skin protection measures. **Methods**: A prospective observational study was carried out. A total of 59 students participated, 32 with a diagnosis of myopia and 27 controls, during SARS-CoV-2 pandemic. Two serological tests for total 25-hydroxy vitamin D (D2 + D3) (Calciferol) were taken, activity habits and sun exposure were identified using the Intermountain Live Well Institute tool, and CUVAF images were taken post-pandemic. **Results**: In the 59 participants, we observed similar vitamin D concentrations between the myopic and control groups during and after the pandemic. However, analysis of CUVAF areas after the pandemic revealed that myopes had significantly smaller areas compared to controls (*p* < 0.05). **Conclusions**: The study demonstrated that using vitamin D as a biomarker for outdoor activity requires additional investigation; the CUVAF biomarker showed a significant association with myopia.

## 1. Introduction

In March 2020, the World Health Organization (WHO) declared the SARS-CoV-2 pandemic. Governments worldwide implemented measures to prevent its spread, focusing primarily on restricting outdoor activities [1,2]. Consequently, the lifestyle of most individuals changed dramatically, with a significant reduction in outdoor exposure time, which increased the risk of myopia progression [3,4].

Scientific evidence demonstrates that reduced outdoor exposure promotes myopia progression by altering intracellular signaling pathways in the eye, involving the retina, choroid, and sclera, leading to excessive eyeball elongation [5,6]. Current objective measures of outdoor time include various methods such as Global Positioning System (GPS) devices with light sensors, serum vitamin D measurement, and conjunctival ultraviolet autofluorescence (CUVAF) area assessment [7,8]. Although GPS devices can detect location changes between indoor and outdoor environments, they do not accurately measure light intensity. Additionally, their functionality depends on continuous use by the individual, proper orientation, and adequate battery life, which limits their application, especially in children [8].

Intrinsic biomarkers, such as vitamin D levels and CUVAF, have been proposed as reliable and viable alternatives for assessing outdoor exposure. Some studies report that the protective effect of increased outdoor exposure is linked to higher vitamin D levels [9]. Vitamin D is synthesized in the skin after exposure to ultraviolet B (UV-B) radiation [10,11,12], and some authors suggest that measuring vitamin D levels can serve as an objective biomarker of outdoor activity [13,14]. However, there is still controversy over whether vitamin D has a protective effect or can solely be considered a biomarker of outdoor activity in myopic patients [15,16]. CUVAF detects preclinical lesions in the bulbar conjunctiva due to exposure to UV radiation caused by the corneal focusing effect of peripheral light, and previous studies have shown a positive correlation between greater sun exposure and larger CUVAF areas [17]. A systematic review and meta-analysis demonstrated an inverse relationship between CUVAF and myopia, validating the utility of CUVAF as a biomarker of outdoor exposure [18]. Moreover, CUVAF area measurement is an objective, simple, rapid, and non-invasive method for assessing outdoor exposure in myopic populations, both children and adults [18,19]. While one report noted that restricted outdoor activity during the pandemic doubled myopia progression among children and young adults [20], information on how SARS-CoV-2 lockdowns affected intrinsic parameters like vitamin D and CUVAF in myopic populations remains limited. This study evaluated the association between serum vitamin D levels and CUVAF area in young adults with myopia during and after the pandemic and their relationship with sun exposure habits and skin protection measures.

## 2. Materials and Methods

### 2.1. Study Design

This prospective observational study was conducted at a higher education institution. A total of 59 university students participated, ranging in age from 16 to 38 years, with 79.7% being female, including 32 diagnosed with myopia and 27 controls. An a priori sample size calculation was performed based on the prevalence of myopia (12.9%) noted in young adults in Colombia as reported in the MIOPUR study [21]. The calculation indicated the need to include 155 participants to achieve 80% power with a 95% confidence level. Participants were recruited during routine clinical consultations at the Optometry Clinic of Antonio Nariño University. The initial selection process was carried out during the period of health restrictions due to the SARS-CoV-2 pandemic, which limited participation among the population due to fear of infection. The exclusion criteria were as follows: (1) documented ophthalmological diseases; (2) anisometropia; (3) patients with avitaminosis; (4) patients taking vitamin D supplements; (5) patients with amblyopia; and (6) students who could not use cycloplegic agents. The control group consisted of emmetropic and hyperopic patients, classified according to refractive errors. The study was approved by the ethics committee of the participating institution under protocol 13012020, in compliance with the Declaration of Helsinki and Colombian regulations on health research (Resolution 8430 of 1993). All participants signed informed consent forms.

### 2.2. Protocol

Refractive error was measured using cycloplegic refraction 30 min after administering one drop each of 0.4% benoxinate, 1% tropicamide, and 1% cyclopentolate. Cycloplegia was confirmed by pupil dilation ≥6 mm and absence of pupillary light reflex.

Refractive error was classified based on spherical equivalent (SE): myopia (≤−0.50 D), hyperopia (≥+2.00 D), and emmetropia (−0.25 D to +1.75 D). All refractive definitions referred to values obtained under cycloplegia [22]. Normal myopia was defined as SE ≤ −0.50 D and ≥−5.75 D, high myopia as SE ≤ −6.00 D, and the control group as SE ≥ −0.50 D. Vitamin D levels were categorized as follows (ng/mL): deficiency (<10), insufficiency (10–30), sufficiency (30–100), toxicity (>100).

Two total 25-hydroxyvitamin D (D2 + D3) serological tests were conducted via phlebotomy following the laboratory protocol. The first test was performed at participants’ residences during outdoor activity restrictions, adhering to international biosafety standards. The second test was conducted at a laboratory one year after the resumption of extracurricular activities. Tests were performed with fasting patients, using polypropylene tubes with silica gel separators. Samples were refrigerated (2–8 °C), protected from light, and transported in hermetically sealed containers with dry ice. Vitamin D levels were measured using chemiluminescence techniques and reported in ng/mL.

Sociodemographic characteristics, activity habits, and sun exposure were identified using an adapted Spanish questionnaire on adolescent health risks, designed by the Intermountain Live Well Institute. This questionnaire integrates lifestyle recommendations from the WHO’s 9th Global Conference on Health Promotion (Shanghai, 2016).

Post-pandemic, CUVAF images were captured using a customized photographic system consisting of a Nikon D5600 DSLR camera and a UV lamp emitting at 365 nm. All images were taken in complete darkness to eliminate ambient visible light that could interfere with the fluorescence signal. The CUVAF area was measured using Fiji/ImageJ software (Java 8.0). For calibration purposes, a 6 mm diameter reference circle (corresponding to a real area of 28.27 mm^2^) was placed on the lower eyelid, 15 mm from the lateral canthus, replicating standard CUVAF imaging conditions. At a fixed distance of 17 cm, images from a sample of 30 eyes were used to calculate a conversion factor to ensure that all CUVAF measurements reflected actual physical units. See conversion formula. Measurements were performed by three evaluators, with intra- and inter-observer reliability assessed.

Conversion factor:
$$\mathrm{Conversion}\;\mathrm{factor}=\frac{\mathrm{Actual}\;\mathrm{area}\;({mm}^{2})}{\mathrm{Fiji}\;\mathrm{area}\;({mm}^{2})}=\frac{28.27\;{mm}^{2}}{316.17\;{mm}^{2}}=0.08942$$

Formula to convert Fiji area to actual area:$$\mathrm{Actual}\;\mathrm{area}\;\left({mm}^{2}\right)=\mathrm{Fiji}\;\mathrm{area}\times 0.08942$$

### 2.3. Statistical Analysis

Continuous variables (e.g., SE, vitamin D) were summarized using mean ± standard deviation (SD); categorical variables were summarized as frequencies and percentages. Normality was assessed using the Kolmogorov–Smirnov test. Differences in sun exposure and skin protection habits between groups were analyzed using Fisher’s exact test and the Mann–Whitney U test for quantitative data. Bivariate analysis included demographic variables and sun exposure/skin care habits over follow-up periods using Fisher’s exact test and odds ratio (OR) calculations with confidence intervals (CI). Two-tailed *t*-tests for paired samples analyzed differences between right and left eyes, nasal and temporal zones (Appendix A), and mean CUVAF area. Pearson’s correlation assessed relationships. Statistical analyses were conducted using SPSS version 25, with *p*-values < 0.05 considered significant.

## 3. Results

Thirty-two myopic students and twenty-seven controls participated in the study. The mean age of participants was 24.97 ± 5.93 years, ranging from 16 to 38 years, and the majority were female (79.7%). Most participants had fair skin color (55.9%) and medium complexion (42.4%). Among the myopic group, 88.1% had low myopia and 11.9% had high myopia. During the pandemic, 56 participants were classified as vitamin D deficient (94.9%), and only three had sufficient levels (5.1%). After the pandemic, 55 participants remained vitamin D deficient, while 4 exhibited sufficient levels (6.8%).

### 3.1. Sun Exposure and Skin Protection Habits

Most young adults in both the myopic and control groups engaged in virtual activities during the pandemic, with a notable increase in in-person work and study activities in the control group after the pandemic. Significant statistical differences were observed between groups regarding occupation mode (*p* = 0.013). However, no significant differences were found between the groups regarding sun exposure and skin care habits during or after the pandemic (Appendix A).

### 3.2. Demographic Characteristics, Skin Care Habits, and Vitamin D Levels

Bivariate analysis revealed a higher risk of vitamin D deficiency in women compared to men after the pandemic. No significant differences were found between study variables and sun exposure or skin care habits, except for using artificial light during the pandemic among young adults with vitamin D sufficiency versus deficiency. Additionally, no statistically significant differences were observed between groups regarding sun exposure duration during or after the pandemic.

The modality of activities or the classification of myopia during or after the pandemic did not show differences between young people with vitamin D insufficiency and sufficiency. During the SARS-CoV-2 pandemic, people who carried out their activities in person had higher levels of Vitamin D between 10 and 30 ng/mL compared to the hybrid and virtual modalities; however, they continued to be classified as insufficient (levels between 20 and 30 ng/mL face-to-face 48%, hybrid 20%, and virtual 32%) (Table 1).

### 3.3. Myopia and Vitamin D Levels During and After the Pandemic

Vitamin D levels showed minimal variation between study groups during and after the pandemic (Table 2).

### 3.4. Myopia and CUVAF Area After the Pandemic

Among the 39 participants evaluated post-pandemic, the mean CUVAF area was 3.94 mm^2^, with significantly smaller areas observed in the myopic group than in controls. When broken down by groups, the mean CUVAF area was 4.76 mm^2^ (±3.2) in the control group, 3.1 mm^2^ (±3.8) in the myopic group, and 1.19 mm^2^ (±1.8) in the high myopia group (Figure 1A). The mean CUVAF area in the high myopia group was significantly smaller than that of the control group (*p* < 0.05) (Figure 1B).

When analyzing the percentage of participants with CUVAF 0 in either the nasal or temporal quadrant of one or both eyes, this percentage was significantly higher in the myopic group compared to controls (*p* < 0.05) (Figure 1C). Furthermore, among myopes, those with high myopia exhibited a significantly greater number of quadrants with CUVAF 0 (*p* < 0.001) (Figure 1D).

Participants with greater myopic refractive errors had a significantly higher proportion of quadrants with CUVAF 0 (Figure 2A). Moreover, a statistically significant positive correlation was identified between spherical equivalent and CUVAF area (Pearson correlation coefficient: r = 0.17; *p* < 0.05), indicating that more negative spherical equivalents were associated with smaller CUVAF areas (Figure 2B). Conversely, no significant correlation was found between CUVAF area and serum vitamin D levels (Pearson correlation coefficient: r = −0.11; *p* = 0.51).

## 4. Discussion

The COVID-19 pandemic posed a global challenge, causing drastic changes in lifestyle within a short period. Since this viral infection spreads through respiratory transmission, the primary prevention strategy involves restrictions on outdoor activities, potentially impacting visual health, particularly in individuals with myopia. Increasing outdoor exposure is one of the most relevant environmental factors for reducing the incidence and progression of myopia [6,23]. Reduced exposure to UV radiation decreases the release of dopamine in the retina, which stimulates axial growth of the eyeball and promotes the progression of myopia [24]. In this context, identifying an objective biomarker for clinical practice is essential. This study evaluated vitamin D levels and CUVAF area as biomarkers of outdoor activity during periods of restricted exposure.

While similar vitamin D concentrations were observed between myopes and controls during and after the pandemic, post-pandemic analysis of CUVAF areas revealed significantly smaller areas in the myopic group than in controls, with the most pronounced differences in the high myopia subgroup. This underscores the association of CUVAF as a biomarker with outdoor activities and myopia. In response to UV radiation, CUVAF records intra- and extracellular changes in the ocular surface in response to UV radiation. The architecture of the conjunctiva contains endogenous fluorophores such as NAD (P)H and flavins FAD, which are involved in cellular energy metabolism, and exogenous fluorophores such as collagen and estatin, a component of the conjunctival extracellular matrix, molecules responsible for conjunctival autofluorescence [20,25]. However, no clear correlation was observed between CUVAF area and serum vitamin D levels.

Although no statistically significant differences in vitamin D levels were identified between myopic and non-myopic individuals during and after the pandemic, the normal myopia group exhibited slightly higher vitamin D levels during both periods (21.35 ng/mL and 21.38 ng/mL, respectively) compared to the high myopia group (17.70 ng/mL and 18.55 ng/mL, respectively). These findings align with previous studies reporting significant differences in serum 25(OH)D levels in children with varying degrees of myopia [26,27,28]. However, it is noteworthy that studies documenting this association primarily involved pediatric populations in growth stages with progressive myopia. In contrast, the current study evaluated an adult population with complete ocular development.

This study also identified a higher risk of vitamin D deficiency in women during and after the pandemic, consistent with previous literature [29,30]. Notably, individuals who predominantly worked or studied under artificial light during the pandemic exhibited a higher risk of vitamin D deficiency than those exposed to natural sunlight. Drastic changes in daily activities during the pandemic, especially in densely populated urban settings, encouraged using poorly illuminated indoor spaces for prolonged study and work activities. However, these preliminary findings do not establish a direct causal relationship between types of lighting and myopia development.

Despite the small sample size, various studies support the independent association between serum vitamin D levels, sun exposure, and myopia in adult populations [31]. While current evidence does not conclusively establish a relationship between vitamin D concentrations and myopia [15,32,33,34], it is well-known that vitamin D synthesis increases with ultraviolet radiation exposure. The exact biological mechanism of vitamin D in myopia development remains unclear [35], but three main hypotheses have been proposed:Changes in intracellular calcium levels may impair ciliary muscle contraction and relaxation, promoting myopia development [35].Serum vitamin D might influence retina–scleral signaling pathways mediated by dopamine and retinoic acid, playing a role in ocular elongation [36].Vitamin D levels could affect scleral metalloproteinase (MMP) activity, influencing ocular morphology and refraction [37].

The COVID-19 lockdown forced students and workers to perform all activities indoors, significantly limiting outdoor activities. This study revealed that most participants had insufficient vitamin D levels (10–30 ng/mL), regardless of refractive error. Vitamin D deficiency has become globally prevalent, even in equatorial countries like Colombia, where high UV radiation exposure would theoretically prevent deficiency. Before the pandemic, vitamin D insufficiency was already highly prevalent in young adults in Colombia, likely due to predominantly indoor activities. Moreover, vitamin D has been suggested as an associative rather than causal factor in various acute and chronic conditions [16,38] which could extend to its relationship with myopia.

Post-pandemic analysis of CUVAF revealed significantly smaller CUVAF areas in the myopic and high-myopia groups than in controls, even in a small sample size. These findings align with recent reports [39,40] confirming that CUVAF is an objective biomarker for monitoring outdoor exposure time in myopic individuals [18]. Consistent with Bilbao et al. [41], this study found a statistically significant correlation between spherical equivalent and CUVAF area. Furthermore, myopic and high-myopia participants had a greater number of quadrants with zero CUVAF area, reflecting that myopic adults primarily engage in academic and occupational activities indoors under artificial lighting, reducing UV radiation exposure. While the effect of ophthalmic protection or sunglasses on CUVAF area was not analyzed in this study, previous evidence suggests that the amount of light reaching the conjunctiva after being filtered by the use of glasses does not significantly alter the size of CUVAF [39,42].

This is the first study to examine CUVAF areas in the Americas, specifically in Colombia, which is located in an equatorial region with consistently high UV radiation. This characteristic may explain why CUVAF areas are larger than countries with lower UV radiation levels throughout the year. Additionally, larger CUVAF areas were observed in the nasal conjunctiva compared to the temporal conjunctiva, consistent with previous studies reporting higher peripheral light incidence on the nasal quadrant compared to the temporal side [18,43].

Although the present study did not identify a significant association between vitamin D levels and the analyzed groups, the findings are similar to those reported by Kearney et al., 2018 [39], and differences in the CUVAF area were observed between individuals with myopia, high myopia, and the control group. However, no statistically significant correlation was found between the CUVAF area and serum vitamin D levels. These findings support CUVAF as an objective, accessible biomarker that is easy to implement in clinical practice, in contrast to vitamin D measurement, a costly, invasive method not routinely employed in optometry and ophthalmology consultations. Human studies are needed to understand how ultraviolet (UV) radiation modulates the activation of vitamin D receptors in the sclera and its involvement in the pathophysiology of myopia. The results of the present study highlight the potential of the CUVAF area as an objective and independent biomarker of UV exposure in individuals with varying degrees of myopia.

One of the main limitations of this study was the sample size due to pandemic-related recruitment challenges. The absence of subsequent refractive data hindered assessing myopia progression, highlighting the need to consider these factors in future research. This study is among the first to examine changes in vitamin D levels during an extended period of restricted outdoor activity in individuals with myopia, and the first to measure CUVAF areas in a region where UV radiation conditions are consistent throughout the year.

## 5. Conclusions

The findings of this study highlight the need for further research with larger sample sizes to more accurately validate the observed associations regarding the use of vitamin D as an indirect biomarker of sunlight exposure, particularly in young adults. However, serum vitamin D measurement has significant clinical limitations, given its invasive nature and high cost. In contrast, conjunctival ultraviolet autofluorescence (CUVAF) is positioned as an objective, accessible, rapid, and cost-effective biomarker that showed a significant association with the presence of myopia, supporting its potential usefulness in monitoring UV radiation exposure and in myopia control strategies.

## Figures and Tables

**Figure 1 biomedicines-13-02042-f001:**
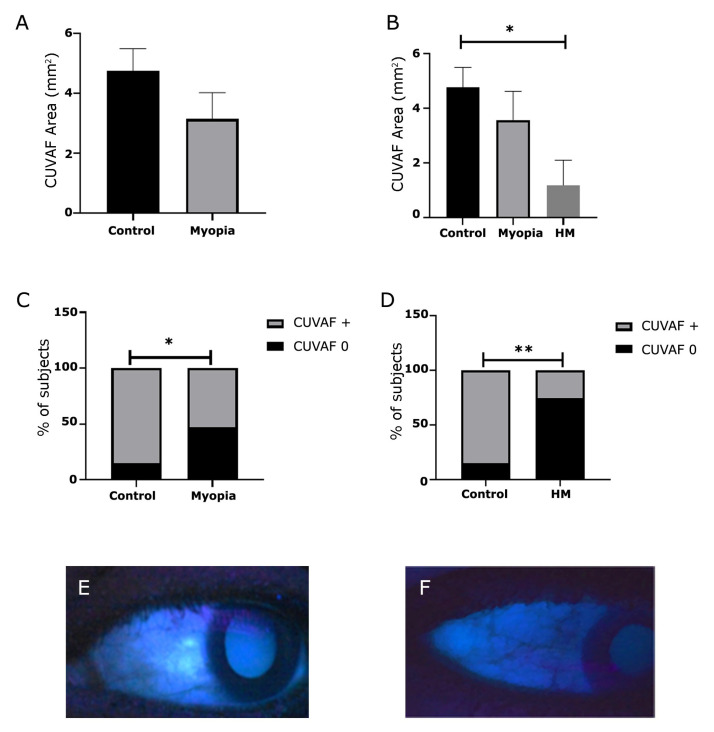
CUVAF area in student population in Colombia. (**A**) Mean CUVAF area differences between controls and myopic group. (**B**) Mean CUVAF area differences between control, myopic, and high myopic groups. (**C**) Differences in the distribution of CUVAF 0 between controls and myopic group. (**D**) Differences in the distribution of CUVAF 0 between controls and high myopia. (**E**) CUVAF + (**F**) CUVAF 0. * *p* < 0.05; ** *p* < 0.01.

**Figure 2 biomedicines-13-02042-f002:**
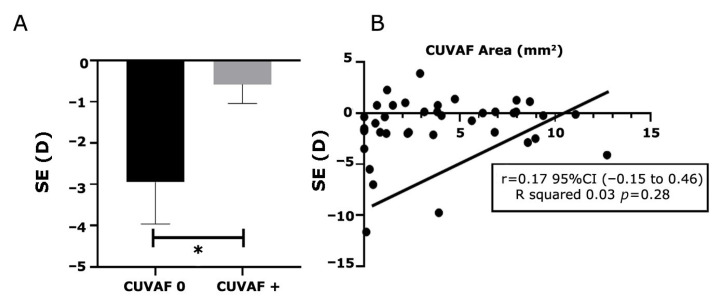
Spherical equivalents and CUVAF correlation. (**A**) Mean differences in SE between participants with CUVAF 0 and positive CUVAF. (**B**) Pearson correlation analysis between CUVAF and spherical equivalent. *: *p* < 0.05.

**Table 1 biomedicines-13-02042-t001:** Bivariate analysis of the characteristics and habits of young Colombians and vitamin D levels during and after the SARS-CoV-2 pandemic. n = 59.

	During the Pandemic				After the Pandemic			
	Insufficient	Sufficient	OR (CI)	*p* Value *	Insufficient	Sufficient	OR (CI)	*p* Value *
Characteristics	n (%)				n (%)			
**Sex**								
Female	46 (97.9)	1 (2.1)	9.2 (0.75–111.62)	0.041	46 (97.9)	1 (2.1)	15.3 (1.42–164.56)	0.005
Male	10 (83.3)	2 (16.7)			9 (75.0)	3 (25.0)		
**Age**								
Under 25 years old	32 (94.1)	2 (5.9)	0.66 (0.05–7.77)	0.615	32 (94.1)	2 (5.9)	1.39 (0.18–10.61)	0.569
From 26 to 38 years old	24 (96.0)	1 (4.0)			23 (92.0)	2 (8.0)		
**Skin Color**								
White	32 (97.0)	1 (3.0)	2.67 (0.22–31.15)	0.578	31 (93.9)	2 (6.1)	1.29 (0.16–9.84)	0.598
Brunette	24 (92.3)	2 (7.7)			24 (92.3)	2 (7.7)		
**Using Sunscreen**								
No	34 (97.1)	1 (2.9)	3.09 (0.26–36.16)	0.359	26 (96.3)	1 (3.7)	2.69 (0.26–27.48)	0.373
Yes	22 (91.7)	2 (8.3)			29 (90.6)	3 (9.4)		
**Using Sunscreen Outdoors**								
No	11 (91.7)	1 (8.3)	0.48 (0.04–5.89)	0.501	10 (90.9)	1 (9.1)	0.66 (0.06–7.09)	0.572
Yes	45 (95.7)	2 (4.3)			45 (93.8)	3 (6.3)		
**Sunscreen Application Zones**								
Face. neck and forearm	12 (92.3)	1 (7.7)	-	0.683	7 (87.5)	1 (12.5)	1.71 (0.09–31.92)	0.715
Face and neck	16 (94.1)	1 (5.9)	-	0.619	19 (95.0)	1 (5.0)	2.28 (0.12–41.98)	0.569
Face	24 (96.0)	1 (4.0)	-	0.683	22 (95.7)	1(4.3)	3.14 (0.17–57.07)	0.418
Do not apply	4 (100)	0	Ref	-	7 (87.5)	1 (12.5)	Ref	-
**Hours of Sun Exposure**								
Less than 2 h	16 (100)	0	1.07 (0.99–1.16)	0.278	20 (95.2)	1 (4.8)	1.71 (0.16–17.60)	0.551
More than 2 h	40 (93.0)	3 (7.0)			35 (92.1)	3 (7.9)		
**Daily Device Usage Time**								
Less than 8 h	31 (93.9)	2 (6.1)	0.62 (0.05–7.24)	0.590	30 (93.8)	2 (6.3)	1.20 (0.15–9.14)	0.627
More than 8 h	25 (96.2)	1 (3.8)			25 (92.6)	2 (7.4)		
**Type of Lighting**								
Artificial	20 (90.9)	2 (9.1)	30 (2.03–441.8)	0.002	19 (90.5)	2 (9.5)	-	0.472
Solar and Artificial	33 (100)	0	-	<0.001	31 (93.9)	2 (6.1)	-	0.571
Solar	3 (75.0)	1 (25.0)	Ref	-	5 (100)	0	Ref	-
**Work/Study Mode**								
Virtual	32 (100)	0	-	0.009	10 (83.3)	2 (16.7)	0.2 (0.01–2.46)	0.173
Hybrid	16 (94.1)	1 (5.9)	4 (0.31–51.02)	0.260	20 (95.2)	1 (4.8)	0.8 (0.04–13.6)	0.877
Presential	8 (80.0)	2 (20.0)	Ref	-	25 (96.2)	1 (3.8)	Ref	-
**Classification of myopia**								
High Myopia	6 (100)	0	-	0.529	6 (100)	0	-	0.491
Normal myopia	24 (92.3)	2 (7.7)	0.46 (0.03–5.42)	0.632	24 (92.3)	2 (7.7)	0.96 (0.12–7.37)	0.968
Control	26 (96.39)	1 (3.7)	Ref	-	25 (96.2)	2 (7.4)	Ref	-

Fischer’s exact test. OR: Odds Ratio. CI: Confidence Interval. *: The cutoff value for vitamin D statistical analysis was set at ≤30 (ng/mL) were defined as insufficiency, whereas levels > 30 ng/mL were considered sufficiency.

**Table 2 biomedicines-13-02042-t002:** Myopia and Vitamin D in young Colombians during and after the SARS-CoV-2 pandemic. n = 59.

	Refractive Data Right Eye	Refractive Data Left Eye	Vitamin D Levels		
			During	After	
	Mean (Min/Max)	Mean (Min/Max)	Mean ± SD	Mean ± SD	*p* Value *
Normal myopia	* −2.35 (−5.75, 1.75)	* −2.08 (−5.00, 1.25)	21.35 ± 6.73	21.38 ± 6.90	0.334
High myopia	* −8.12 (−12.00, −3.25)	* −7.50 (−13.00, −2.75)	17.70 ± 7.17	18.55 ± 5.18	0.463
Control	* 0.91 (−0.50, 5.00)	* 1.16 (−0.25, 4.75)	20.35 ± 5.86	20.29 ± 5.18	0.859
Vitamin D levels all groups	* −1.44 (−12.00, 5.00)	* −1.14 (−13.00, 4.75)	20.52 ± 6.36	20.59 ± 6.23	0.739

* Paired *t*-test. SD: Standard Deviation.

## Data Availability

Due to data protection, the data is not on any platform, but the provision of data to requesters will be considered.

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
