# Peer review of "Objective Biomarkers of Outdoor Activity (Vitamin D and CUVAF) in Young Adults with Myopia During and After the SARS-CoV-2 Pandemic"

_biomedicines, 2025, doi:10.3390/biomedicines13082042_

Round 1
Reviewer 1 Report
Comments and Suggestions for Authors
This article primarily investigates the association between objective biomarkers of outdoor activities (specifically, serum vitamin D levels and conjunctival ultraviolet autofluorescence area, CUVAF) and myopia in young people during and after the SARS-CoV-2 pandemic. The manuscript is generally well-organized with a clear structure, guiding the reader through the background, methods, results, and conclusions. However, given that numerous studies have already explored the correlations between serum vitamin D levels, CUVAF, and outdoor activities, despite the inclusion of the SARS-CoV-2 factor in this study, its research value is significantly constrained. Furthermore, the small sample size of this study further limits its significance. Here are some suggestions:
- The methods section is comprehensive but would benefit from a more detailed description of the participant recruitment process, including how participants were approached and response rates, loss to follow-up rates.
- The use of a custom photography system for CUVAF image capture was innovative. However, providing more details about the calibration process and the reliability of the device under different lighting conditions can enhance the reproducibility of your study.
- The statistical analysis is robust, but consider including a power calculation to demonstrate that your sample size was adequate to detect meaningful differences.
- The discussion effectively interprets the results in the context of existing literature. However, consider expanding on the potential mechanisms underlying the observed associations between CUVAF area and myopia. For instance, discuss how reduced UV exposure might affect ocular development and myopia progression at a cellular or molecular level.
Author Response
First, I would like to sincerely thank you for the substantial efforts you have made in revising the manuscript. In response to your comments, we have made the necessary adjustments in the indicated sections.
- The methods section is comprehensive but would benefit from a more detailed description of the participant recruitment process, including how participants were approached and response rates, loss to follow-up rates.
Comment 1
Sample recruitment is included in the study design. Lines 84-87
Study Design
This prospective observational study was conducted at a higher education institution. A total of 59 university students participated, ranging in age from 16 to 38 years, with 79.7% being female,, including 32 diagnosed with myopia and 27 controls. . An a priori sample size calculation was performed based on the prevalence of myopia (12.9%) noted in young adults in Colombia as reported in the MIOPUR study (1) . The calculation indicated the need to include 155 participants to achieve 80% power with a 95% confidence level. Participants were recruited during routine clinical consultations at the Optometry Clinic of Antonio Nariño University. The initial selection process was carried out during the period of health restrictions due to the SARS-CoV-2 pandemic, which limited participation among the population due to fear of infection. Exclusion criteria were: 1) documented ophthalmological diseases; 2) anisometropia; 3) patients with avitaminosis; 4) patients taking vitamin D supplements; 5) patients with amblyopia; and 6) students unable to use cycloplegic agents. The control group consisted of emmetropic and hyperopic patients, classified based on refractive errors. The study was approved by the ethics committee of Universidad Antonio Nariño under protocol 13012020, adhering to the Helsinki Declaration and Colombian health research regulations (Resolution 8430 of 1993). All participants signed informed consent forms.
- The use of a custom photography system for CUVAF image capture was innovative. However, providing more details about the calibration process and the reliability of the device under different lighting conditions can enhance the reproducibility of your study.
Comment 2
The calibration process is included in the protocol section. Line 139 - 146
Protocol
After the COVID-19 pandemic, CUVAF images were captured using a customized photographic system consisting of a Nikon D5600 DSLR camera and a UV lamp emitting at 365 nm. All images were taken in complete darkness to eliminate ambient visible light that could interfere with the fluorescence signal. The CUVAF area was measured using Fiji/ImageJ software. For calibration purposes, a 6 mm diameter reference circle (corresponding to a real area of 28.27 mm²) was placed on the lower eyelid, 15 mm from the lateral canthus, replicating standard CUVAF imaging conditions.At a fixed distance of 17 cm, images from a sample of 30 eyes were used to calculate a conversion factor to ensure that all CUVAF measurements reflected actual physical units. See conversion formula. All measurements were performed by three independent evaluators, and both intra- and inter-observer reliability were assessed.
- The statistical analysis is robust, but consider including a power calculation to demonstrate that your sample size was adequate to detect meaningful differences.
Comment 3
A sample calculation is included in the study design section. Lines 80-84
This prospective observational study was conducted at a higher education institution. A total of 59 university students participated, ranging in age from 16 to 38 years, with 79.7% being female,, including 32 diagnosed with myopia and 27 controls. . An a priori sample size calculation was performed based on the prevalence of myopia (12.9%) noted in young adults in Colombia as reported in the MIOPUR study. The calculation indicated the need to include 155 participants to achieve 80% power with a 95% confidence level. (1)Participants were recruited during routine clinical consultations at the Optometry Clinic of Antonio Nariño University. The initial selection process was carried out during the period of health restrictions due to the SARS-CoV-2 pandemic, which limited participation among the population due to fear of infection. Exclusion criteria were: 1) documented ophthalmological diseases; 2) anisometropia; 3) patients with avitaminosis; 4) patients taking vitamin D supplements; 5) patients with amblyopia; and 6) students unable to use cycloplegic agents. The control group consisted of emmetropic and hyperopic patients, classified based on refractive errors. The study was approved by the ethics committee of Universidad Antonio Nariño under protocol 13012020, adhering to the Helsinki Declaration and Colombian health research regulations (Resolution 8430 of 1993). All participants signed informed consent forms.
- The discussion effectively interprets the results in the context of existing literature. However, consider expanding on the potential mechanisms underlying the observed associations between CUVAF area and myopia. For instance, discuss how reduced UV exposure might affect ocular development and myopia progression at a cellular or molecular level.
Comment 4
The discussion section includes the associations observed between the CUVAF area and myopia and how reduced exposure to UV rays could affect eye development and the progression of myopia at the cellular or molecular level. Lines 165-267 y 274-279
Discussion
The COVID-19 pandemic posed a global challenge, causing drastic changes in lifestyle within a short period. Since this viral infection spreads through respiratory transmission, the main prevention strategy involved restrictions on outdoor activities, potentially impacting visual health, particularly in individuals with myopia. Increasing outdoor exposure is one of the most relevant environmental factors for reducing the incidence and progression of myopia (2,3). Reduced exposure to UV radiation decreases the release of dopamine in the retina, which stimulates axial growth of the eyeball and promotes the progression of myopia (4).In this context, identifying an objective biomarker for clinical practice is essential. This study evaluated vitamin D levels and CUVAF area as biomarkers of outdoor activity during periods of restricted exposure.
While similar vitamin D concentrations were observed between myopes and controls during and after the pandemic, post-pandemic analysis of CUVAF areas revealed significantly smaller areas in the myopic group compared to controls, with the most pronounced differences seen in the high myopia subgroup. This underscores the association of CUVAF as a biomarker with outdoor activities and myopia. In response to UV radiation, CUVAF records intra- and extracellular changes in the ocular surface in response to UV radiation. The architecture of the conjunctiva contains endogenous fluorophores such as NAD (P)H and flavins FAD, which are involved in cellular energy metabolism, and exogenous fluorophores such as collagen and estatin, a component of the conjunctival extracellular matrix, molecules responsible for conjunctival autofluorescence (5)(6).However, no clear correlation was observed between CUVAF area and serum vitamin D levels.

Reviewer 2 Report
Comments and Suggestions for Authors
Comments:
- I don’t see any promising association between CUVAF biomarker and myopia.
- The sample size is too small to draw meaningful conclusions.
- Line 110-113- I am not sure if paragraph “Normal myopia was defined as SE ≤ -0.50D and ≥ -5.75D, high myopia as SE ≤ -6.00D, and the control group as SE ≥ -0.50D. Continuous variables (e.g., SE, vitamin D) were summarized using mean ± standard deviation (SD); categorical variables were summarized as frequencies and percentages. Vitamin D levels were categorized as follows (ng/mL): deficiency (<10), insufficiency (10–30), sufficiency (30–100), toxicity (>100)” can be a part of statistical analysis. May be try rephasing or shift somewhere else.
- I suggest, rewriting the methods section as it should have contained detailed subjects’ information like age, sex etc.
- Did wearing sunglasses instead of just applying sunscreen on skin, affect CUVAF biomarker and myopia in these patients?
- Were there no error bars on fig 1, C and D? and what does % on y axis reflects, please mention full name.
Overall nice preliminary findings which require further studies to be more conclusive.

Author Response
First, I would like to sincerely thank you for the substantial efforts you have made in revising the manuscript. In response to your comments, we have made the necessary adjustments in the indicated sections.
- I don’t see any promising association between CUVAF biomarker and myopia.
Comment 1
In the introduction and discussion section, we expand on why CUVAF is a biomarker of outdoor activity in myopia.Line 64-71, 264-266 and 273-278
- Introduction
Intrinsic biomarkers, such as vitamin D levels and CUVAF, have been proposed as reliable and viable alternatives for assessing outdoor exposure. Some studies report that the protective effect of increased outdoor exposure is linked to higher vitamin D levels (7). Vitamin D is synthesized in the skin after exposure to ultraviolet B (UV-B) radiation (8–10), and some authors suggest that measuring vitamin D levels can serve as an objective biomarker of outdoor activity (11,12). However, there is still controversy over whether vitamin D has a protective effect or can solely be considered a biomarker of outdoor activity in myopic patients (13,14). CUVAF detects preclinical lesions in the bulbar conjunctiva due to exposure to UV radiation caused by the corneal focusing effect of peripheral light, and previous studies have shown a positive correlation between greater sun exposure and larger CUVAF areas.(15) .A systematic review and meta-analysis demonstrated an inverse relationship between CUVAF and myopia, validating the utility of CUVAF as a biomarker of outdoor exposure (16). Moreover, CUVAF area measurement is an objective, simple, rapid, and non-invasive method for assessing outdoor exposure in myopic populations, both children and adults (16,17). While one report noted that restricted outdoor activity during the pandemic doubled myopia progression among children and young adults (18), information on how SARS-CoV-2 lockdowns affected intrinsic parameters like vitamin D and CUVAF in myopic populations remains limited. Therefore, this study aimed to evaluate the association between serum vitamin D levels and CUVAF area in young adults with myopia during and after the pandemic, as well as their relationship with sun exposure habits and skin protection measures.
Discussion
The COVID-19 pandemic posed a global challenge, causing drastic changes in lifestyle within a short period. Since this viral infection spreads through respiratory transmission, the main prevention strategy involved restrictions on outdoor activities, potentially impacting visual health, particularly in individuals with myopia. Increasing outdoor exposure is one of the most relevant environmental factors for reducing the incidence and progression of myopia (2,3). Reduced exposure to UV radiation decreases the release of dopamine in the retina, which stimulates axial growth of the eyeball and promotes the progression of myopia (4).In this context, identifying an objective biomarker for clinical practice is essential. This study evaluated vitamin D levels and CUVAF area as biomarkers of outdoor activity during periods of restricted exposure.
While similar vitamin D concentrations were observed between myopes and controls during and after the pandemic, post-pandemic analysis of CUVAF areas revealed significantly smaller areas in the myopic group compared to controls, with the most pronounced differences seen in the high myopia subgroup. This underscores the association of CUVAF as a biomarker with outdoor activities and myopia. In response to UV radiation, CUVAF records intra- and extracellular changes in the ocular surface in response to UV radiation. The architecture of the conjunctiva contains endogenous fluorophores such as NAD (P)H and flavins FAD, which are involved in cellular energy metabolism, and exogenous fluorophores such as collagen and estatin, a component of the conjunctival extracellular matrix, molecules responsible for conjunctival autofluorescence (5)(6).However, no clear correlation was observed between CUVAF area and serum vitamin D levels.
- The sample size is too small to draw meaningful conclusions.
Comment 2
In the limitations and conclusions section, we explain the difficulties of sample size associated with the restrictions of the Sarcov 2 pandemic. Lines 378-380 and 385-391
Discussion
One of the main limitations of this study was the sample size due to pandemic-related recruitment challenges. The absence of subsequent refractive data hindered assessing myopia progression, highlighting the need to consider these factors in future research. This study is among the first to examine changes in vitamin D levels during an extended period of restricted outdoor activity in individuals with myopia, and the first to measure CUVAF areas in a region where UV radiation conditions are consistent throughout the year
Conclusions
The findings of this study highlight the need for further research with larger sample sizes to more accurately validate the observed associations regarding the use of vitamin D as an indirect biomarker of sunlight exposure, particularly in young adults. However, serum vitamin D measurement has significant clinical limitations, given its invasive nature and high cost. In contrast, conjunctival ultraviolet autofluorescence (CUVAF) is positioned as an objective, accessible, rapid, and cost-effective biomarker that showed a significant association with the presence of myopia, supporting its potential usefulness in monitoring UV radiation exposure and in myopia control strategies.
- Line 110-113- I am not sure if paragraph “Normal myopia was defined as SE ≤ -0.50D and ≥ -5.75D, high myopia as SE ≤ -6.00D, and the control group as SE ≥ -0.50D. Continuous variables (e.g., SE, vitamin D) were summarized using mean ± standard deviation (SD); categorical variables were summarized as frequencies and percentages. Vitamin D levels were categorized as follows (ng/mL): deficiency (<10), insufficiency (10–30), sufficiency (30–100), toxicity (>100)” can be a part of statistical analysis. May be try rephasing or shift somewhere else.
Comment 3
The parameters for defining myopia are included in the protocol section. Lines 123-126
Protocol
Refractive error was measured using cycloplegic refraction 30 minutes after administering one drop each of 0.4% benoxinate, 1% tropicamide, and 1% cyclopentolate. Cycloplegia was confirmed by pupil dilation ≥6 mm and absence of pupillary light reflex.
Refractive error was classified based on spherical equivalent (SE): myopia (≤ -0.50D), hyperopia (≥ +2.00D), and emmetropia (-0.25D to +1.75D). All refractive definitions referred to values obtained under cycloplegia (19). Normal myopia was defined as SE ≤ -0.50D and ≥ -5.75D, high myopia as SE ≤ -6.00D (21), and the control group as SE ≥ -0.50D. Vitamin D levels were categorized as follows (ng/mL): deficiency (<10), insufficiency (10–30), sufficiency (30–100), toxicity (>100).
The statistical analysis was limited to aspects specifically related to this section. Lines 150-175
Statistical Analysis
Continuous variables (e.g., SE, vitamin D) were summarized using mean ± standard deviation (SD); categorical variables were summarized as frequencies and percentages. Normality was assessed using the Kolmogorov-Smirnov test. Differences in sun exposure and skin protection habits between groups were analyzed using Fisher's exact test and the Mann-Whitney U test for quantitative data. Bivariate analysis included demographic variables and sun exposure/skin care habits over follow-up periods using Fisher's exact test and odds ratio (OR) calculations with confidence intervals (CI). Two-tailed t-tests for paired samples analyzed differences between right and left eyes, nasal and temporal zones, and mean CUVAF area. Pearson's correlation assessed relationships. Statistical analyses were conducted using SPSS version 25, with p-values <0.05 considered significant.
- I suggest, rewriting the methods section as it should have contained detailed subjects’ information like age, sex etc.
Comment 4
In the study design, we included the characteristics of the population. Lines 81-82
Study Design
This prospective observational study was conducted at a higher education institution. A total of 59 university students participated, ranging in age from 16 to 38 years, with 79.7% being female,, including 32 diagnosed with myopia and 27 controls. An a priori sample size calculation was performed based on the prevalence of myopia (12.9%) noted in young adults in Colombia as reported in the MIOPUR study. The calculation indicated the need to include 155 participants to achieve 80% power with a 95% confidence level. (1)Participants were recruited during routine clinical consultations at the Optometry Clinic of Antonio Nariño University. The initial selection process was carried out during the period of health restrictions due to the SARS-CoV-2 pandemic, which limited participation among the population due to fear of infection.. Exclusion criteria were: 1) documented ophthalmological diseases; 2) anisometropia; 3) patients with avitaminosis; 4) patients taking vitamin D supplements; 5) patients with amblyopia; and 6) students unable to use cycloplegic agents. The control group consisted of emmetropic and hyperopic patients, classified based on refractive errors. The study was approved by the ethics committee of Universidad Antonio Nariño under protocol 13012020, adhering to the Helsinki Declaration and Colombian health research regulations (Resolution 8430 of 1993). All participants signed informed consent forms.
- Did wearing sunglasses instead of just applying sunscreen on skin, affect CUVAF biomarker and myopia in these patients?
Comment 5
In the discussion section, we expand on the effect of optical correction on the CUVAF area. Lines 334-337.
Discussion
While the effect of ophthalmic protection or sunglasses on CUVAF area was not analyzed in this study, previous evidence suggests that the amount of light reaching the conjunctiva after being filtered by the use of glasses does not significantly alter the size of CUVAF (20,21).
.
- Were there no error bars on fig 1, C and D? and what does % on y axis reflects, please mention full name.
Comment 6
We have modified Figure 1 to clarify that the Y-axis represents the percentage of cases (% of subjects) with either positive or negative CUVAF in the different groups: Control vs Myopia (Figure 1C) and Control vs High Myopia (Figure 1D). Since this is a frequency analysis, the data reflect proportions of individuals in each category, and therefore no error bars are included, as there is no standard deviation applicable in this type of categorical comparison.
Overall nice preliminary findings which require further studies to be more conclusive.

Round 2
Reviewer 1 Report
Comments and Suggestions for Authors
The quality of descriptions in the methodology, results, and discussion sections of this paper has significantly improved after revisions. However, there are still two critical issues in this study that cannot be resolved through word optimization:
- Insufficient sample size. Based on the prevalence rate of myopia in young adults in Colombia, the study estimated that 155 participants needed to be recruited to ensure 80% statistical power and a 95% confidence level. However, due to recruitment difficulties during the pandemic, only 59 individuals were included (comprising 32 myopia patients and 27 controls). This situation has multiple negative impacts: On one hand, it may lead to a reduction in the study's statistical power, making it possible that the study fails to detect differences or associations even when they truly exist, thereby increasing the risk of false negatives. On the other hand, a small sample size is prone to causing bias in estimates, undermining the authenticity and reliability of the results.
- Issues regarding research innovativeness. Currently, numerous studies have explored the correlations among serum vitamin D levels, CUVAF (continuous ultraviolet-A free environment), and outdoor activities. Although this study was conducted during and after the SARS-CoV-2 pandemic, it has not demonstrated significant innovativeness.
Author Response
We appreciate your comments and observations on the manuscript. Taking into account the observations made on the sample size, we consider:
Comment 1
- Insufficient sample size. Based on the prevalence rate of myopia in young adults in Colombia, the study estimated that 155 participants needed to be recruited to ensure 80% statistical power and a 95% confidence level. However, due to recruitment difficulties during the pandemic, only 59 individuals were included (comprising 32 myopia patients and 27 controls). This situation has multiple negative impacts: On one hand, it may lead to a reduction in the study's statistical power, making it possible that the study fails to detect differences or associations even when they truly exist, thereby increasing the risk of false negatives. On the other hand, a small sample size is prone to causing bias in estimates, undermining the authenticity and reliability of the results. Lines 284-286. Reference 45.
Although the present study did not identify a significant association between vitamin D levels and the analyzed groups he findings are similar to those reported by Kearney et al, 2018 [45].
Thank you for your comment. We acknowledge that this study's main limitation is the small sample size
Previous studies, such as that of Kearney et al, 2018, [1], with a sample of 54 participants (24 myopic and 30 non-myopic), have reported consistent findings, showing a negative association between conjunctival ultraviolet autofluorescence area (CUVAF) and the presence of myopia, reporting nonsignificant association between sun exposure or serum vitamin D levels and refractive error, which is consistent with some of the findings obtained in the present study. It should be noted that, due to the restrictions imposed by the COVID-19 pandemic, it was impossible to reach the sample size initially planned. We agree on the need to develop future studies with analytical and longitudinal designs with larger sample sizes and more robust statistical power to reinforce the evidence reported in this study, minimize possible association biases, and establish causal relationships with greater robustness. Nevertheless, we consider that the results are encouraging and reinforce the potential of CUVAF as an objective biomarker of exposure to ultraviolet radiation in different geographical locations.
Comment 2
Issues regarding research innovativeness. Currently, numerous studies have explored the correlations among serum vitamin D levels, CUVAF (continuous ultraviolet-A free environment), and outdoor activities. Although this study was conducted during and after the SARS-CoV-2 pandemic, it has not demonstrated significant innovativeness. Lines 277-278
This is the first study to examine CUVAF areas in the Americas, specifically in Colombia is located in an equatorial region with consistently high UV radiation.
Thank you for your comment. This study is the first analysis conducted in Latin America to explore the association between conjunctival ultraviolet autofluorescence area (CUVAF), serum vitamin D levels, and the diagnosis of myopia. Colombia, due to its geographical location close to the equator and its combination of latitude and altitude, has high levels of ultraviolet (UV) radiation, significantly differing from other regions of the world. These characteristics make the country an ideal setting for investigating the relationship between sun exposure, skin synthesis of vitamin D, and it’s possible links to the development and progression of myopia. In this context, the present study contributes significantly to the knowledge of the environmental factors involved in the etiology of myopia in populations living in tropical areas, providing local evidence that could guide specific prevention and control strategies for this geographical region.
Reference
Kearney, S.; O’Donoghue, L.; Pourshahidi, L.K.; Richardson, P.; Laird, E.; Healy, M.; Saunders, K.J. Conjunctival Ultraviolet Autofluorescence Area, but Not Intensity, Is Associated with Myopia. Clin Exp Optom 2018, 102, 43–50, doi:10.1111/cxo.12825.

Reviewer 2 Report
Comments and Suggestions for Authors
I am not fully convinced with this research in its current form , even though author made improvements . I still think, there should be more experiments to strengthen the findings such as exploring the mechanistic part of UV radiation on these individuals. Vitamin D pathways like vitamin D receptors in regulating scleral growth leading to myopia should be explored.
Author Response
We appreciate your comments and observations on the manuscript.
Comment 1
I am not fully convinced with this research in its current form , even though author made improvements I still think, there should be more experiments to strengthen the findings such as exploring the mechanistic part of UV radiation on these individuals. Vitamin D pathways like vitamin D receptors in regulating scleral growth leading to myopia should be explored. Lines 291-295
Human studies are needed to understand how ultraviolet (UV) radiation modulates the activation of vitamin D receptors in the sclera and its involvement in the pathophysiology of myopia. The results of the present study highlight the potential of the CUVAF area as an objective and independent biomarker of UV exposure in individuals with varying degrees of myopia.
We strongly agree that scientific evidence supports the role of vitamin D receptor activation in the sclera, which increases collagen expression and could modulate myopia progression through this cellular pathway. In the present study, an analytical approach of environmental factors mediated by sun exposure and time spent outdoors with serum 25-hydroxyvitamin D and the presence of myopia in a population living in a geographic region such as Colombia, close to the equator, which makes it a tropical country with high levels of UV radiation. Our results can be considered a gateway to understand better the cellular bases that modulate the association between serum vitamin D levels and myopia, mainly in contexts with the previously described characteristics.

Round 3
Reviewer 1 Report
Comments and Suggestions for Authors
After revision, the article has reached the level of publication.
Author Response
We appreciate your comments and observations on the manuscript.
Reviewer 2 Report
Comments and Suggestions for Authors
Thankyou for providing clarifications and improving the manuscript.
I would recommend to add a line in abstract background (as currently no background info was provided in the abstract), if any association is reported on myopia, vitamin D and CoV-2.
No further comments at my end.
Author Response
We appreciate your comments and observations on the manuscript.
Comment 1
I would recommend to add a line in abstract background (as currently no background info was provided in the abstract), if any association is reported on myopia, vitamin D and CoV-2. Lines 20-25.
Intrinsic biomarkers, such as serum vitamin D levels and conjunctival ultraviolet autofluorescence area (CUVAF), have been proposed to quantify sunlight exposure. Evidence suggests that reduced outdoor activity during the SARS-CoV-2 pandemic accelerated the progression of myopia; however, there is little information on the impact of such restrictions on vitamin D levels and CUVAF area in populations with myopia.
